# WaveletDiff: Multilevel Wavelet Diffusion For Time Series Generation

## Abstract

Time series are ubiquitous in many applications that involve forecasting, classification and causal inference tasks, such as healthcare, finance, audio signal processing and climate sciences. Still, large, high-quality time series datasets remain scarce. Synthetic generation can address this limitation; however, current models confined either to the time or frequency domains struggle to reproduce the inherently multi-scaled structure of real-world time series. We introduce WaveletDiff, a novel framework that trains diffusion models *directly on wavelet coefficients* to exploit the inherent multi-resolution structure of time series data. The model combines dedicated transformers for each decomposition level with cross-level attention mechanisms that enable selective information exchange between temporal and frequency scales through adaptive gating. It also incorporates energy preservation constraints for individual levels based on Parseval's theorem to preserve spectral fidelity throughout the diffusion process. Comprehensive tests across six real-world datasets from energy, finance, and neuroscience domains demonstrate that WaveletDiff consistently outperforms state-of-the-art time-domain and frequency-domain generative methods on both short and long time series across five diverse performance metrics. For example, WaveletDiff achieves discriminative scores and Context-FID scores that are $3\times$ smaller on average than the second-best baseline across all datasets.

## 1 Introduction

Time series data arises in diverse practical settings, including healthcare Lee et al. (2019a); van der Schaar Lab (2019), finance Sezer et al. (2020); Özbayoğlu et al. (2020), climate sciences Dinku (2019); Climate Change AI (2021), audio processing Mitra & Zualkernan (2025) and engineering Susto et al. (2020); Lei et al. (2020); Carvalho et al. (2022). Due to various constraints, acquiring sufficiently high-quality labeled time-series datasets remains a challenge Wang et al. (2021); Desai et al. (2025). The problem may be mitigated through synthetic time series generation, which also offers promising solutions for data augmentation Wen et al. (2021); Le Guennec et al. (2018); Ryu et al. (2023), privacy preservation Wang et al. (2020); Jordon et al. (2024); Nosowsky & Giordano (2006), forecasting Taga et al. (2025) and simulations Nikolenko (2021); El Emam et al. (2022).

Current time series generation methods predominantly operate either directly in the time domain or frequency domain, and come with different advantages and limitations. Time-domain approaches, including those based on GANs Yoon et al. (2019); Pei et al. (2021); EskandariNasab et al. (2024), autoregressive Salinas et al. (2020) and diffusion models Lim et al. (2023); Narasimhan et al. (2024); Sikder et al. (2024) are well-suited for modeling local temporal patterns, but struggle with long-term dependencies and preservation of important spectral characteristics. To address time-domain induced limitations, recent approaches have increasingly leveraged frequency-domain analysis, often along with temporal modeling Tian et al. (2020); Chi et al. (2024); Crabbé et al. (2024); Huang et al. (2024). These methods are of relevance since many real-world time series tend to exhibit higher localization in the frequency rather than the time domain. Representative methods include FourierFlow Alaa et al. (2021), which applies normalizing flows to Fourier representations, DiffusionTS Yuan & Qiao (2024), which combines Fourier decompositions with diffusion models, and various frequency-enhanced transformers Zhou et al. (2022a;b); Xu et al. (2024); Yi et al. (2023) that use both spectral and temporal analyses. However, these approaches typically process time and frequency domain information either in a separate manner or impose trade-offs between temporal

resolution and spectral coherence. They are also not able to simultaneously capture both local and global time and spectral patterns, which is crucial for synthesizing realistic time series.

Wavelet transforms represent a natural approach to address the above issue by creating a multi-resolution representation that simultaneously captures both temporal and spectral information Mallat (1989); Cohen (2001). Unlike the Fourier transform, which capture global frequency properties, wavelets maintain temporal localization while also providing useful decompositions into multiple frequency bands Rioul & Flandrin (1992); Daubechies (1988). This results in a highly versatile time-frequency hierarchical representation Mallat (1989). As a result, wavelet-based analyses have been used with success for various signal processing applications including speech recognition, financial trends analysis, image processing, and biomedical signal analysis Daubechies (1992); Vetterli & Kovačević (1995); Burrus et al. (1998). Despite these results, a handful of known wavelet-based approaches for time series generation have failed to provide improvements over Fourier-based methods Takahashi & Mizuno (2024); Kazemi & Meidani (2022); Zhao et al. (2018). This may be attributed to the fact that, almost exclusively, the methods treat wavelet coefficients as image structures and then follow-up by applying standard image generation techniques such as convolutional neural networks or image-based diffusion models. While potentially useful for data-poor applications and highly specialized time series, indirect time series → wavelet → image conversion methods in general suffer from pattern distortions caused by noninvertible image features.

A more adequate approach based on wavelet decompositions is to run diffusion models directly in the wavelet domain, which is a new direction proposed in this work. For diffusion, methods such as denoising diffusion probabilistic models (DDPMs) Ho et al. (2020) which have demonstrated remarkable success in image Dhariwal & Nichol (2021), audio Kong et al. (2021), and text generation Austin et al. (2021), are considered state-of-the-art for time-series generation. However, these diffusion models are tailor-made for highly specific time-series formats (e.g., audio or financial data), and may not be suitable for other modalities. This motivates us to implement a new wavelet-space diffusion model, termed WaveletDiff, which is universally applicable as it inherently respects different multi-level structures. Unlike frequency-domain approaches, WaveletDiff also captures temporal patterns at different scales simultaneously. Our key innovations lie in running forward diffusion processes for each wavelet level individually and in parallel, following fine-tuned exponential noise mechanisms and using dedicated level-transformer denoising networks combined with a cross-level attention mechanism that enables information exchange between different decomposition levels. This design preserves the hierarchical nature of wavelet representations while allowing the model to learn complex inter-scale dependencies crucial for realistic time series generation.

Our technical contributions include:
**1.** A diffusion framework that operates directly in the wavelet domain and tunes the noise addition process to the approximation and detail levels, identifies the most suitable choice of mother wavelet for different time series and uses level-specific loss functions and transformers.
**2.** A cross-scale attention mechanisms that enables information flow between different temporal scales while preserving their individual properties.
**3.** A wavelet-aware loss weighting mechanism that prevents some levels from dominating the training objective through level-specific balancing strategies.
**4.** A new evaluation metric based on the Dynamic Time-Warping distance and extensive comparative analysis of both short and long time series datasets, including ETTh1, ETTh2, Stocks, Exchange Rate, fMRI, EEG Zhou et al. (2021); Lai et al. (2018); Roesler (2013). The results show significant performance gains of WaveletDiff compared to Fourier and time-based methods, which are roughly three-fold on average for discriminative and Context-FID scores.
**5.** The first empirical evaluation of reproducibility of diffusion models for time series, akin to recent efforts reported for images Zhang et al. (2024b); Li et al. (2024); Kadkhodaie et al. (2024).

## 2 RELATED WORK

**Time Series Generation with Diffusion Models.** Early generative AI methods for time series focused on conditional generation tasks such as forecasting and imputation Rasul et al. (2021); Tashiro et al. (2021); Li et al. (2022); Yang et al. (2024), while recent approaches target unconditional time series generation Shen & Kwok (2023); Barancikova et al. (2025). The above methods employ various architectural choices including RNNs, transformers, and specialized denoising networks to handle the sequential nature of temporal data Kong et al. (2021), and almost exclusively operate in the time domain. This limits their ability to capture global and local spectral properties.

**Frequency Domain Approaches for Time Series.** Recent works have demonstrated that real-world time series are more localized in the frequency domain, making spectral diffusion more effective than time diffusion Crabbé et al. (2024). Various frequency-based approaches include lightweight models using complex-valued operations Xu et al. (2024), frequency-enhanced transformers combining discrete Fourier transform (DFT) with attention mechanisms Zhou et al. (2022b;a), and specialized MLP architectures for frequency learning Yi et al. (2023). Additional methods incorporate spectral filtering Zhang et al. (2024a), multi-resolution frequency analysis Wang et al. (2024), and normalizing flows in Fourier domain Alaa et al. (2021). Additional unconditional generation approaches include interpretable diffusion models that combine *trend and seasonality* components Yuan & Qiao (2024) and latent diffusion models that operate in compressed latent spaces for more efficient generation Qian et al. (2024). These methods often require separate processing pipelines for temporal and frequency components, limiting their ability to simultaneously capture multi-scale temporal-spectral relationships.

**Wavelet-Based Time Series Modeling.** Wavelet transforms provide multi-resolution time-frequency representation capabilities Mallat (1989); Daubechies (1992); Addison (2017) and have been extensively used in time series analysis Percival & Walden (2000); Sang (2013); Patrik et al. (2015). In generative modeling, wavelets have shown promise across various domains through direct coefficient processing Phung et al. (2022); Hu et al. (2023); Guth et al. (2022). For time series forecasting, wavelets have been employed to enhance traditional forecasting models through their multi-resolution time-frequency analysis capabilities Zhou et al. (2025); Sasal et al. (2022); Arabi et al. (2024); Schlüter & Deuschle (2010). For generation tasks, existing methods predominantly convert wavelet coefficients to image representations for processing with standard computer vision techniques Takahashi & Mizuno (2024); Kazemi & Meidani (2022). However, this indirect approach may not fully exploit the hierarchical multi-scale structure of wavelet decompositions, where each level captures distinct temporal and spectral characteristics.

## 3 METHODOLOGY

### 3.1 WAVELET REPRESENTATIONS OF TIME SERIES

A multivariate time series dataset $\mathbf{X} \in \mathbb{R}^{N \times T \times D}$ with $N$ samples, $T$ timesteps and $D$ features (e.g., opening price, closing price, high/low, volume for financial data) comprises time series of the form $\mathbf{x}^{(i)} = [\mathbf{x}_0^{(i)}, \mathbf{x}_1^{(i)}, \dots, \mathbf{x}_{T-1}^{(i)}] \in \mathbb{R}^{T \times D}$, where $i \in [1, N]$. The Discrete Wavelet Transform (DWT) decomposes each time series through a cascade of high-pass and low-pass filtering operations followed by downsampling. The decomposition utilizes a scaling function $\phi(t)$ and its associated *mother wavelet* $\psi(t)$. The mother wavelet is characterized by its order $p$, which ensures that the wavelet is orthogonal to all polynomials of degree less than $p$ (hence, $p$ determines the number of vanishing moments). Higher-order wavelets provide better frequency localization but require longer filters. The filter length $F$ represents the number of nonzero coefficients in the discrete filters, which depends on the wavelet family and order $p$ (e.g., $F = 2p$ for Daubechies wavelets). More details are available in Appendix A. These functions satisfy the two-scale relations:

$$\psi(t) = \sqrt{2} \sum_{k=0}^{F-1} g_k \phi(2t - k), \text{ where } \phi(t) \text{ satisfies } \phi(t) = \sqrt{2} \sum_{k=0}^{F-1} h_k \phi(2t - k), \quad (1)$$

and where $\{g_k\}_{k=0}^{F-1}$ and $\{h_k\}_{k=0}^{F-1}$ are the high-pass and low-pass filter coefficients, respectively, with the relationship $g_k = (-1)^k h_{F-1-k}$ ensuring orthogonality. The DWT performs recursive decomposition over $L$ levels. Starting with the approximation coefficients $\mathbf{A}^{(0)} = \mathbf{X}$, at each level $l \in [1, L]$, we apply high-pass and low-pass filters followed by temporal-dimension downsampling:

$$\boldsymbol{C}_{:,m,:}^{(l)} = \sum_k g_k \, \boldsymbol{A}_{:,2m-k,:}^{(l-1)} \text{ (detail coeff.)}, \quad \boldsymbol{A}_{:,m,:}^{(l)} = \sum_k h_k \, \boldsymbol{A}_{:,2m-k,:}^{(l-1)} \text{ (approximate coeff.)}, \quad (2)$$

where $m$ indexes the downsampled time dimension and the operation is applied independently across all $N$ samples and $D$ features. Boundary effects are handled using symmetric extension, where the signal is mirrored at the tails to ensure sufficient coefficients for filtering operations. This decomposition yields the wavelet coefficient representation:

$$\text{DWT}(\mathbf{X}) = \{\boldsymbol{C}^{(1)}, \dots, \boldsymbol{C}^{(L)}, \boldsymbol{A}^{(L)}\}, \quad (3)$$

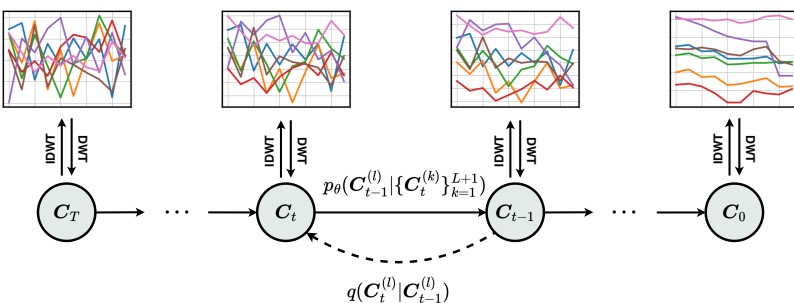

Figure 1: Direct wavelet coefficient diffusion, where the forward process proceeds independently at each decomposition level, while the reverse process integrates information across all levels to perform collective denoising.

where $\boldsymbol{C}^{(l)} \in \mathbb{R}^{N \times d_l \times D}$, for $l \in [1, L]$, are the detail coefficients and $\boldsymbol{A}^{(L)} \in \mathbb{R}^{N \times d_L \times D}$ are the approximation coefficients. For consistency with diffusion notation, we write $\boldsymbol{C}^{(L+1)} = \boldsymbol{A}^{(L)}$.

The wavelet order $p$ is chosen based on the sequence length to ensure sufficient coefficients at each level, with longer sequences accommodating higher-order wavelets for better frequency localization. The coefficient dimension at each level $l$ is calculated recursively as:

$$d_l = \lfloor \frac{d_{l-1} + F - 1}{2} \rfloor, \quad l = 1, \dots, L, \tag{4}$$

where $d_0 = T$ is the original sequence length. This formula accounts for the filter overlap (requiring $F - 1$ additional boundary coefficients) and dyadic downsampling (division by 2) inherent to the wavelet decomposition process. The number of decomposition levels $L$ is determined based on the sequence length $T$ to ensure sufficiently coefficients are available at each level while maintaining meaningful frequency separation, and is set to $L = \max\left(3, \min\left(7, \lfloor \log_2\left(\frac{T}{F-1}\right) \rfloor\right)\right)$ in practice.

To reconstruct time series from diffusion-generated wavelet coefficients $\{\hat{\boldsymbol{C}}^{(1)}, \dots, \hat{\boldsymbol{C}}^{(L)}, \hat{\boldsymbol{C}}^{(L+1)}\}$, we apply the Inverse Discrete Wavelet Transform (IDWT). The reconstruction proceeds from the coarsest level to the finest level. Starting with $\hat{\boldsymbol{A}}^{(L)} = \hat{\boldsymbol{C}}^{(L+1)}$, for each $l = L, \dots, 1$, we compute:

$$\hat{\boldsymbol{A}}^{(l-1)}_{:,m,:} = \sum_k \tilde{h}_{m-2k} \hat{\boldsymbol{A}}^{(l)}_{:,k,:} + \sum_k \tilde{g}_{m-2k} \hat{\boldsymbol{C}}^{(l)}_{:,k,:}, \tag{5}$$

where $\tilde{h}$ and $\tilde{g}$ are the synthesis filters used for reconstruction. For orthogonal wavelets, these take the form $\tilde{h}_k = h_{-k}$ and $\tilde{g}_k = g_{-k}$, while for biorthogonal wavelets, they are independently designed dual filters that ensure perfect reconstruction. The reconstruction combines the current approximation $\hat{\boldsymbol{A}}^{(l)}$ with the detail coefficients $\hat{\boldsymbol{C}}^{(l)}$ through upsampling and filtering. The inverse transform can hence be written as:

$$\hat{\mathbf{X}} = \text{IDWT}(\{\hat{\boldsymbol{C}}^{(1)}, \dots, \hat{\boldsymbol{C}}^{(L)}, \hat{\boldsymbol{A}}^{(L)}\}), \tag{6}$$

where $\hat{\mathbf{X}} = \hat{\boldsymbol{A}}^{(0)} \in \mathbb{R}^{N \times T \times D}$ is the reconstructed time series.

### 3.2 WAVELET-SPACE DIFFUSION FRAMEWORK

We propose to run the diffusion process on wavelet coefficients using Denoising Diffusion Probabilistic Models (DDPM), as shown in Figure 1. The forward diffusion process in the wavelet domain gradually adds Gaussian noise to coefficients at all levels independently:

$$q(\boldsymbol{C}^{(l)}_t | \boldsymbol{C}^{(l)}_0) = \mathcal{N}(\boldsymbol{C}^{(l)}_t; \sqrt{\bar{\alpha}_t}\boldsymbol{C}^{(l)}_0, (1 - \bar{\alpha}_t)\mathbf{I}), \quad l = 1, \dots, L+1, \tag{7}$$

where $\alpha_t = 1 - \beta_t$ and $\bar{\alpha}_t = \prod_{s=1}^t \alpha_s$ follow standard DDPM schedules. Specifically, we adopt an exponential noise schedule $\beta_t = \beta_{start} + (\beta_{end} - \beta_{start}) \cdot (1 - e^{-\gamma \cdot t})$, where $\gamma$ is the exponential decay rate, $t \in [0, 1]$ is the normalized timestep, and $\beta_{start}$ and $\beta_{end}$ are tuneable hyperparamters. Exponential schedules are better suited for wavelet-based time series generation than cosine schedules. This may be because cosine schedules start slowly, peak mid-epoch, and then decrease, with

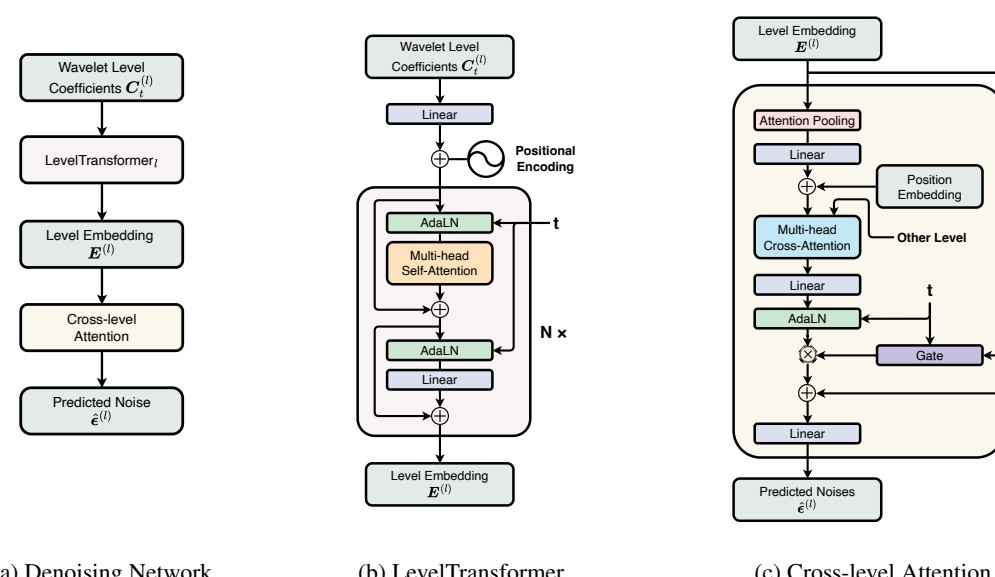

(a) Denoising Network    (b) LevelTransformer    (c) Cross-level Attention

Figure 2: The wavelet coefficients are independently processed by LevelTransformers to obtain level-specific embeddings. These embeddings are obtained through interaction across levels via a cross-level attention module based on adaptive gating mechanisms.

this smooth behavior stabilizing high-dimensional data like images. In contrast, time series and their wavelet decompositions have significantly lower dimensions, benefiting from more aggressive noise injection when coupled with transformer-based denoising models used at early backwards steps.

We parameterize the reverse process using a cross-level transformer network that employs cross-attention to enable communication across levels:

$$p_\theta(\boldsymbol{C}_{t-1}^{(l)}|\{\boldsymbol{C}_t^{(k)}\}_{k=1}^{L+1}) = \mathcal{N}(\boldsymbol{C}_{t-1}^{(l)}; \boldsymbol{\mu}_\theta(\{\boldsymbol{C}_t^{(k)}\}_{k=1}^{L+1}, t), \boldsymbol{\Sigma}_\theta), \quad l = 1, \dots, L+1, \tag{8}$$

where $\boldsymbol{\mu}_\theta(\{\boldsymbol{C}_t^{(k)}\}_{k=1}^{L+1}, t)$ represents the predicted mean of the reverse diffusion process, parameterized by the neural network $\theta$ and conditioned on all wavelet levels and the diffusion timestep $t$, and $\boldsymbol{\Sigma}_\theta$ is the predicted covariance matrix. Following the DDPM framework, we fix $\boldsymbol{\Sigma}_\theta = \beta_t \mathbf{I}$ and train the denoising network $f_\theta$ to predict the added noise $\boldsymbol{\epsilon}$ using the mean square error (MSE) as the loss:

$$\hat{\boldsymbol{\epsilon}}^{(l)} = f_\theta(\{\boldsymbol{C}_t^{(k)}\}_{k=1}^{L+1}, t), \quad l = 1, \dots, L+1 \tag{9}$$

$$\mathcal{L}_{\text{recon}} = \mathbb{E}_{\boldsymbol{C}, t, \epsilon} \left[ \sum_{l=1}^{L+1} w_l \cdot \|\boldsymbol{\epsilon}^{(l)} - \hat{\boldsymbol{\epsilon}}^{(l)}\|^2 \right] \tag{10}$$

where $\boldsymbol{\epsilon}^{(l)}$ and $\hat{\boldsymbol{\epsilon}}^{(l)}$ are the true and predicted noise at level $l$, and $w_l$ are level-specific weights ensuring balanced contribution across scales.

To preserve data spectra for long sequence generation, we optionally introduce an energy conservation penalty based on Parseval's theorem:

$$\mathcal{L}_{\text{energy}} = \mathbb{E}_{\boldsymbol{C}, t, \epsilon} \left[ \sum_{l=1}^{L+1} \left| \mathcal{E}^{(l)} - \hat{\mathcal{E}}^{(l)} \right| \right], \tag{11}$$

where $\mathcal{E}^{(l)} = \sum_{n=1}^{N} \sum_{j=1}^{d_l} \sum_{k=1}^{D} (\boldsymbol{C}_{n,j,k}^{(l)})^2$ represents the true energy at wavelet level $l$, and $\hat{\mathcal{E}}^{(l)} = \sum_{n=1}^{N} \sum_{j=1}^{d_l} \sum_{k=1}^{D} (\hat{\boldsymbol{C}}_{n,j,k}^{(l)})^2$ represents the predicted energy at wavelet level $l$. By enforcing energy preservation at each decomposition level individually, the constraints stabilize training and preserve the natural energy distribution across frequency scales. The overall training objective combines both the reconstruction and energy terms $\mathcal{L} = \mathcal{L}_{\text{recon}} + \lambda_{\text{energy}} \mathcal{L}_{\text{energy}}$, where $\lambda_{\text{energy}}$ denotes the weight of the energy loss term. For short sequences, the base reconstruction loss is typically sufficient since

the spectral energy drift is minimal over limited temporal horizons. The energy preservation term mostly benefits datasets with strong low-frequency trends and smooth spectral characteristics (e.g., ETTh1, Exchange Rate), while high-volatility datasets with abrupt changes (e.g., Stocks) are better reproduced through the reconstruction loss alone.

The denoising network $f_\theta$ uses dedicated transformers for each wavelet level. We adopt Adaptive Layer Normalization (AdaLN) Peebles & Xie (2022) as the normalization layer. For level $l$, the coefficients $C_t^{(l)}$ are processed through a specialized transformer,

$$\mathbf{E}^{(l)} = \text{LevelTransformer}_l(C_t^{(l)}, \mathbf{t}), \quad l = 1, \ldots, L+1, \tag{12}$$

where $\mathbf{t}$ denotes the diffusion time embedding and $\mathbf{E}^{(l)} \in \mathbb{R}^{N \times h_l \times D}$ represents the output level embeddings for level $l$, with $h_l$ denoting the embedding dimension at that level. The embeddings of each level are aggregated through attention-based pooling. Cross-level attention operates on these aggregated representations, allowing each level to adaptively incorporate contextual information from other scales through learned gating mechanisms (Figure 2).

## 4 EXPERIMENTS

### 4.1 EXPERIMENTAL SETTINGS

**Benchmarks**   We compare WaveletDiff to several state-of-the-art time series generation methods, including FourierDiffusion Crabbé et al. (2024), Diffusion-TS Yuan & Qiao (2024), TimeGAN Yoon et al. (2019), and SigDiffusion Barancikova et al. (2025).

**Datasets**   We use six real-world datasets to evaluate our method, covering energy, finance, and neuroscience domains. **ETTh1** and **ETTh2** Zhou et al. (2021) are electricity transformer datasets containing oil temperature and six power load features recorded hourly from 2016 to 2018. **Stocks** is a multivariate financial time series dataset containing historical Google stock market data with price and volume features from 2004 to 2019. **Exchange Rate** Lai et al. (2018) contains daily exchange rates of eight countries from 1990 to 2016. **fMRI** is the NetSim dataset containing simulated BOLD time series data for evaluating network modeling methods in functional magnetic resonance imaging. **EEG** Roesler (2013) contains multichannel electroencephalogram recordings that measure brain electrical activity over time, offering information about neural dynamics and cognitive processes. For more details, refer to Appendix B.1.

**Metrics**   We evaluate generation quality using five complementary metrics. The **discriminative score** measures similarity between real and generated samples by training a binary classifier to distinguish them Yoon et al. (2019). The **predictive score** assesses the utility of synthetic data for forecasting real sequences using mean absolute error. **Context-Fréchet inception distance (Context-FID)** Paul et al. (2022) quantifies distributional distance using TS2Vec Yue et al. (2022) embeddings following Yuan & Qiao (2024). The **correlational score** evaluates temporal dependencies by comparing cross-correlation matrices. Additionally, we propose a novel metric, termed the **Dynamic Time Warping Jensen-Shannon Distance (DTW-JS distance)**, which combines the temporal alignment features of DTW with the JS divergence for distributional comparison. DTW aims to capture optimal temporal alignments between two time series sequences $x$ and $y$ by minimizing

$$\text{DTW}(x, y) = \min_\pi \sum_{(i,j) \in \pi} |x_i - y_j|, \tag{13}$$

where $\pi$ is a warping path allowing flexible temporal matching. We first create a reference set $\mathcal{M}$ by randomly sampling sequences from the union of the real $\mathcal{R}$ and generated $\mathcal{G}$ datasets, which are matched in size. For each sequence $s$ in the real dataset, we compute its mean DTW distance to all sequences in the reference set: $d_\mathcal{R}(s) = \frac{1}{|\mathcal{M}|} \sum_{r \in \mathcal{M}} \text{DTW}(s, r)$. We perform the same calculation for each generated sequence to obtain $d_\mathcal{G}(s)$. This creates two collections of mean distances across all choices of $s$, which we convert into empirical distributions $D_\mathcal{R}$ and $D_\mathcal{G}$. We then apply Jensen-Shannon divergence to compare these distance distributions:

$$\text{DTW-JS}(D_\mathcal{R}, D_\mathcal{G}) = \frac{1}{2}[\text{KL}(D_\mathcal{R}||D_M) + \text{KL}(D_\mathcal{G}||D_M)] \tag{14}$$

where $D_M = \frac{1}{2}(D_{\mathcal{R}} + D_{\mathcal{G}})$ is the mixture of the two distance distributions. Small DTW-JS values indicate that the real and generated samples are "distributionally" similar in terms of their temporal patterns. More details can be found in Appendix B.2.

## 4.2 SHORT SEQUENCE TIME SERIES GENERATION

We follow the evaluation setup of TimeGAN Yoon et al. (2019) and Diffusion-TS Yuan & Qiao (2024) to assess generation quality against baseline models. All datasets are segmented into sequences of length 24 using a sliding window with stride 1. For evaluation, we generate samples matching the size of the original training data for each dataset to ensure fair evaluation. Training configurations and times, as well as model complexity are discussed in Appendices B.4 and B.5.

Table 1: Time series generation performance comparison on short sequences (length 24).

| Metric | Methods | ETTh1 | ETTh2 | Stocks | Exchange Rate | fMRI | EEG |
|---|---|---|---|---|---|---|---|
| Discriminative Score (Lower the Better) | WaveletDiff | **0.005±.005** | **0.008±.007** | **0.005±.004** | **0.004±.001** | **0.087±.077** | **0.006±.008** |
| | FourierDiffusion | 0.019±.007 | 0.016±.006 | 0.024±.003 | 0.015±.009 | 0.196±.013 | 0.016±.007 |
| | Diffusion-TS | 0.071±.002 | 0.038±.008 | 0.087±.008 | 0.032±.002 | 0.188±.018 | 0.304±.177 |
| | TimeGAN | 0.127±.047 | 0.106±.035 | 0.091±.047 | 0.257±.070 | 0.499±.001 | 0.161±.063 |
| | SigDiffusions | 0.353±.023 | 0.381±.048 | 0.371±.027 | 0.324±.055 | 0.482±.018 | 0.500±.000 |
| Predictive Score (Lower the Better) | WaveletDiff | **0.119±.002** | **0.106±.004** | **0.037±.000** | 0.037±.002 | **0.100±.000** | **0.000±.000** |
| | FourierDiffusion | 0.120±.005 | 0.111±.003 | **0.037±.000** | 0.040±.001 | **0.100±.000** | **0.000±.000** |
| | Diffusion-TS | 0.120±.004 | 0.107±.003 | **0.037±.000** | **0.037±.002** | **0.100±.000** | 0.001±.000 |
| | TimeGAN | 0.152±.015 | 0.128±.005 | 0.038±.000 | 0.064±.005 | 0.124±.002 | **0.000±.000** |
| | SigDiffusions | 0.131±.002 | 0.125±.003 | 0.040±.001 | 0.089±.006 | 0.105±.000 | **0.000±.000** |
| Context-FID Score (Lower the Better) | WaveletDiff | **0.020±.001** | **0.023±.002** | **0.018±.002** | **0.006±.000** | **0.104±.006** | **0.006±.000** |
| | FourierDiffusion | 0.031±.002 | 0.024±.003 | 0.093±.010 | 0.054±.013 | 0.169±.005 | 0.012±.001 |
| | Diffusion-TS | 0.151±.007 | 0.054±.002 | 0.187±.016 | 0.056±.007 | 0.106±.003 | 0.017±.001 |
| | TimeGAN | 0.661±.041 | 0.157±.011 | 0.110±.012 | 0.660±.042 | 1.404±.114 | 0.018±.001 |
| | SigDiffusions | 2.413±.179 | 1.053±.099 | 3.494±.383 | 1.691±.157 | 6.576±.210 | 0.022±.001 |
| Correlational Score (Lower the Better) | WaveletDiff | **0.043±.008** | **0.083±.016** | **0.005±.003** | **0.060±.020** | **1.177±.031** | **1.811±.963** |
| | FourierDiffusion | 0.046±.009 | 0.095±.016 | 0.013±.003 | 0.072±.019 | 1.184±.023 | 3.544±.626 |
| | Diffusion-TS | 0.051±.007 | 0.089±.022 | 0.009±.007 | 0.115±.016 | 1.382±.036 | 4.764±.107 |
| | TimeGAN | 0.202±.010 | 0.185±.015 | 0.053±.003 | 0.416±.018 | 29.562±.067 | 8.820±.121 |
| | SigDiffusions | 0.210±.010 | 0.430±.025 | 0.070±.005 | 0.943±.024 | 15.389±.064 | 4.389±.257 |
| DTW-JS distance (Lower the Better) | WaveletDiff | **0.101±.016** | **0.064±.014** | **0.106±.027** | **0.121±.029** | **0.191±.043** | **0.055±.011** |
| | FourierDiffusion | 0.105±.022 | 0.073±.014 | 0.138±.024 | 0.130±.021 | 0.286±.042 | 0.067±.017 |
| | Diffusion-TS | 0.111±.020 | 0.087±.012 | 0.153±.019 | 0.139±.031 | 0.237±.042 | 0.220±.013 |
| | TimeGAN | 0.155±.030 | 0.097±.042 | 0.142±.028 | 0.231±.025 | 0.215±.037 | 0.632±.049 |
| | SigDiffusions | 0.259±.024 | 0.273±.034 | 0.377±.068 | 0.376±.036 | 0.693±.000 | 0.293±.125 |

As shown in Table 1, our method consistently outperforms all baseline methods across all datasets and metrics. While FourierDiffusion achieves competitive performance on certain datasets, WaveletDiff demonstrates superior and more consistent results across all evaluation scenarios. Notably, it achieves $3\times$ lower discriminative and Context-FID scores on average than the second-best baseline across all datasets. In our evaluations, we also tested different wavelet families and selected Symlets wavelet for Stocks, Coeiflets wavelets for fMRI, and Daubechies wavelets for other datasets. In general, even when universally adopting Daubechies wavelets, WaveletDiff outperforms other diffusion paradigms. More details regarding the influence of the wavelet basis function on generative performance are available in Appendix B.3. Furthermore, unlike methods such as DiffusionTS that require nonuniversal trend and seasonality decompositions, our wavelet approach automatically identifies time-frequency patterns. The cross-level attention mechanism also allows for reconstructing fine-grained temporal details while maintaining global spectral coherence. This is particularly evident in terms of consistent improvements of Context-FID scores, which measure distributional similarity using learned temporal representations.

To highlight our model's ability to capture real data distributions, we present t-SNE embeddings and probability density plots for ETTh1 and Stocks in Figures 3 and 4, respectively (see also Appendix C.1). The density plots reveal near-perfect alignment between real and generated distributions, outperforming FourierDiffusion and Diffusion-TS.

## 4.3 LONG TIME SERIES GENERATION

We assess the performance of WaveletDiff for long time series generation by segmenting datasets into sequences of length 32, 64, and 128, again using a sliding window with stride 1. The same evaluation protocol is applied, where we generate samples matching the size of the original training

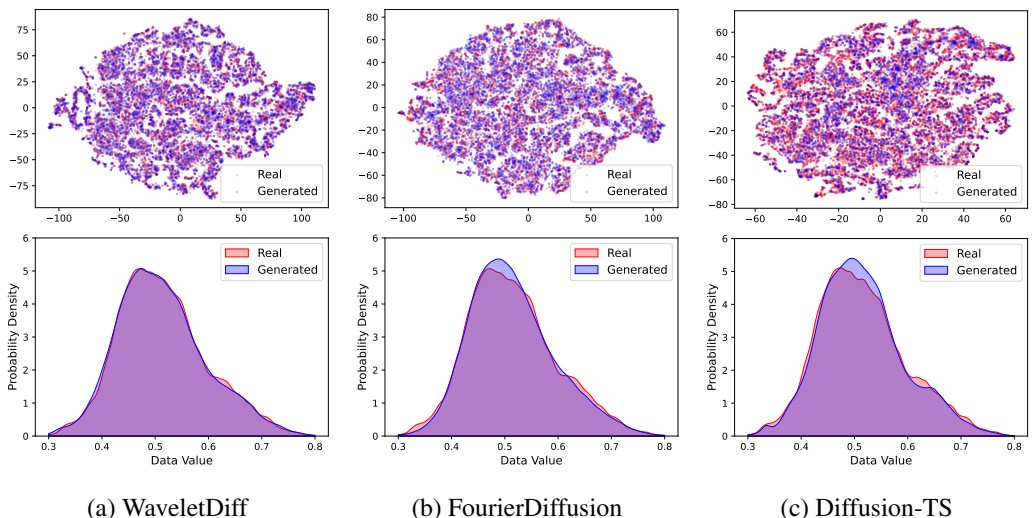

Figure 3: t-SNE visualization and probability distributions of training/synthetic data for ETTh1.

data for each dataset. As shown in Table 2, WaveletDiff once again offers consistent performance improvements across all settings. Here we used the spectral energy preservation term based on Parseval's theorem with a loss weight $\lambda_{\text{energy}} = 0.3$ on ETTh1 and Exchange Rate, but not on Stocks due to its high volatility.

## 4.4 ABLATION STUDY

To validate the effectiveness of different architectural components of WaveletDiff, we conduct an ablation study comparing our full model against four variants: (1) *Predicting coefficients rather than noise:* Instead of predicting noise as in standard DDPM, this variant directly predicts the wavelet coefficients themselves, following the approach in Diffusion-TS Yuan & Qiao (2024) which suggests this method outperforms noise prediction. (2) *Removing cross-attention:* This variant disables information exchange between different wavelet decomposition levels, but maintains level-specific transformer models. (3) *Using Cosine instead of Exponential schedules:* This variant replaces exponential with cosine noise scheduling during diffusion. (4) *Using DDIM sampling:* This variant uses deterministic DDIM rather than DDPM sampling during inference. The results in Table 7 in the Appendix reveal that *cross-level attention* is the universally most critical architectural component.

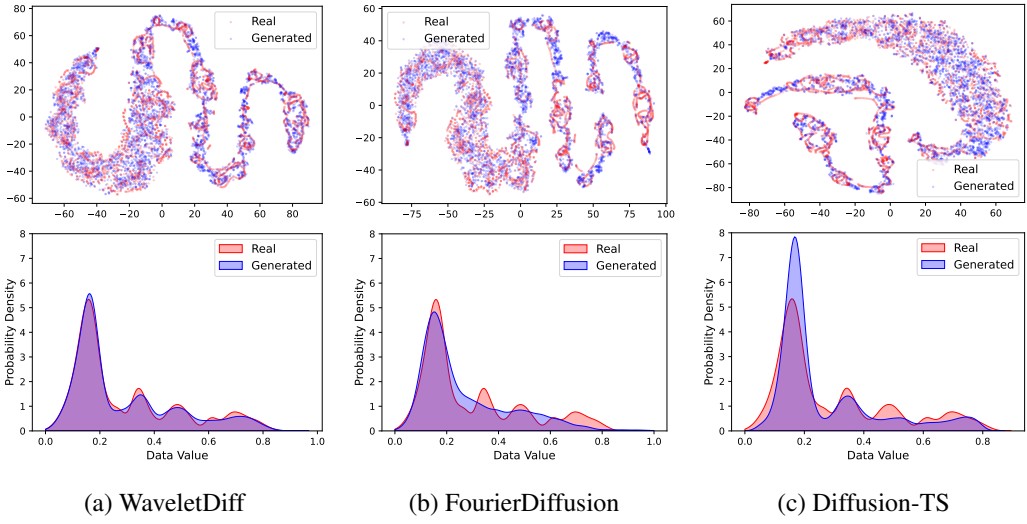

Figure 4: t-SNE visualization and probability distributions of training/synthetic data for Stocks.

Table 2: Time series generation performance comparison on long sequences.

| Dataset | Metric | Length | WaveletDiff | FourierDiffusion | Diffusion-TS | TimeGAN | SigDiffusions |
|---------|--------|--------|-------------|------------------|--------------|---------|---------------|
| ETTh1 | Discriminative | 32 | **0.016±.001** | 0.030±.004 | 0.078±.003 | 0.128±.036 | 0.346±.033 |
| | Score | 64 | **0.028±.009** | 0.048±.004 | 0.079±.010 | 0.116±.088 | 0.294±.156 |
| | (Lower the Better) | 128 | **0.034±.037** | 0.113±.006 | 0.159±.006 | 0.299±.148 | 0.462±.035 |
| | Predictive | 32 | **0.119±.001** | **0.119±.005** | **0.119±.003** | 0.126±.009 | 0.129±.000 |
| | Score | 64 | **0.114±.007** | **0.114±.004** | 0.120±.004 | 0.125±.004 | 0.129±.002 |
| | (Lower the Better) | 128 | 0.113±.005 | **0.112±.007** | 0.116±.005 | 0.177±.015 | 0.129±.002 |
| | Context-FID | 32 | **0.038±.005** | 0.048±.003 | 0.204±.011 | 0.599±.044 | 2.875±.027 |
| | Score | 64 | **0.088±.005** | 0.135±.010 | 0.265±.012 | 0.978±.114 | 6.622±.354 |
| | (Lower the Better) | 128 | **0.256±.014** | 0.356±.021 | 0.805±.094 | 11.813±.851 | 11.596±.800 |
| | Correlational | 32 | **0.050±.004** | 0.056±.019 | 0.064±.014 | 0.118±.013 | 0.180±.013 |
| | Score | 64 | 0.054±.009 | **0.052±.006** | 0.059±.010 | 0.307±.015 | 0.200±.023 |
| | (Lower the Better) | 128 | **0.059±.021** | 0.072±.010 | 0.083±.004 | 1.098±.005 | 0.235±.015 |
| | DTW-JS | 32 | **0.095±.022** | 0.099±.035 | 0.113±.022 | 0.226±.019 | 0.235±.017 |
| | Distance | 64 | **0.095±.028** | 0.105±.031 | 0.123±.034 | 0.208±.017 | 0.199±.031 |
| | (Lower the Better) | 128 | 0.105±.035 | 0.134±.022 | 0.122±.017 | 0.262±.051 | **0.122±.021** |
| Stocks | Discriminative | 32 | **0.006±.004** | 0.022±.012 | 0.099±.012 | 0.197±.025 | 0.357±.027 |
| | Score | 64 | **0.007±.003** | 0.032±.018 | 0.099±.008 | 0.152±.020 | 0.324±.044 |
| | (Lower the Better) | 128 | **0.015±.008** | 0.086±.036 | 0.141±.011 | 0.270±.124 | 0.339±.007 |
| | Predictive | 32 | **0.037±.000** | **0.037±.000** | 0.038±.000 | **0.037±.000** | 0.040±.001 |
| | Score | 64 | **0.036±.000** | **0.036±.000** | 0.037±.000 | 0.038±.000 | 0.039±.000 |
| | (Lower the Better) | 128 | **0.036±.000** | 0.038±.000 | 0.037±.000 | 0.070±.007 | 0.040±.000 |
| | Context-FID | 32 | **0.026±.006** | 0.087±.007 | 0.256±.029 | 0.449±.042 | 3.403±.373 |
| | Score | 64 | **0.047±.005** | 0.151±.026 | 0.369±.065 | 0.336±.046 | 4.229±.495 |
| | (Lower the Better) | 128 | **0.080±.012** | 0.379±.025 | 0.417±.077 | 3.231±.325 | 5.472±.004 |
| | Correlational | 32 | **0.002±.002** | 0.011±.001 | 0.017±.007 | 0.094±.006 | 0.075±.004 |
| | Score | 64 | **0.003±.001** | 0.013±.005 | 0.020±.002 | 0.098±.003 | 0.052±.004 |
| | (Lower the Better) | 128 | **0.004±.002** | 0.162±.011 | 0.021±.006 | 0.621±.006 | 0.091±.004 |
| | DTW-JS | 32 | **0.112±.025** | 0.118±.021 | 0.137±.026 | 0.182±.026 | 0.301±.060 |
| | Distance | 64 | 0.136±.021 | 0.139±.008 | **0.136±.018** | 0.155±.031 | 0.261±.013 |
| | (Lower the Better) | 128 | 0.112±.013 | 0.127±.020 | **0.116±.004** | 0.420±.015 | 0.281±.058 |
| Exchange Rate | Discriminative | 32 | **0.011±.005** | 0.018±.013 | 0.031±.006 | 0.254±.064 | 0.314±.024 |
| | Score | 64 | **0.020±.005** | 0.038±.015 | 0.028±.005 | 0.277±.046 | 0.300±.007 |
| | (Lower the Better) | 128 | **0.026±.008** | 0.092±.032 | 0.046±.007 | 0.106±.064 | 0.276±.015 |
| | Predictive | 32 | **0.035±.002** | 0.040±.002 | 0.036±.002 | 0.069±.006 | 0.085±.007 |
| | Score | 64 | **0.035±.001** | 0.041±.001 | **0.035±.002** | 0.056±.005 | 0.078±.008 |
| | (Lower the Better) | 128 | **0.034±.003** | 0.044±.002 | **0.034±.002** | 0.048±.003 | 0.074±.005 |
| | Context-FID | 32 | **0.013±.001** | 4.057±.648 | 0.037±.003 | 1.038±.144 | 1.853±.164 |
| | Score | 64 | **0.022±.003** | 0.129±.065 | 0.056±.005 | 1.136±.114 | 1.834±.235 |
| | (Lower the Better) | 128 | **0.052±.003** | 0.264±.008 | 0.063±.004 | 0.849±.087 | 2.079±.168 |
| | Correlational | 32 | **0.064±.010** | 0.109±.037 | 0.091±.044 | 0.456±.016 | 1.065±.033 |
| | Score | 64 | **0.066±.024** | 0.096±.026 | 0.097±.012 | 0.421±.038 | 1.042±.027 |
| | (Lower the Better) | 128 | **0.065±.026** | 0.173±.011 | 0.101±.019 | 0.237±.035 | 1.001±.046 |
| | DTW-JS | 32 | **0.108±.037** | 0.116±.028 | 0.129±.023 | 0.182±.023 | 0.375±.045 |
| | Distance | 64 | **0.132±.015** | 0.142±.028 | 0.136±.010 | 0.195±.036 | 0.305±.024 |
| | (Lower the Better) | 128 | **0.124±.018** | 0.146±.029 | 0.145±.028 | 0.224±.020 | 0.306±.022 |

With its removal causing discriminative scores and Context-FID scores to degrade on average by approximately $4\times$ and $3.5\times$, respectively. While cosine noise scheduling and DDIM sampling show competitive performance on certain datasets, they exhibit instability in neuroscience domain datasets fMRI and EEG. Additionally, our analysis confirms that coefficient prediction consistently underperforms the standard DDPM noise prediction paradigm across all datasets and metrics, in contrast to the findings of Diffusion-TS which is applied directly to the time domain.

**Reproducibility Analysis.** Inspired by recent diffusion model reproducibility studies Zhang et al. (2024b); Li et al. (2024); Kadkhodaie et al. (2024), we examine whether this phenomenon extends to time series generation. To this end, we train model pairs with architectural variations, generate samples from identical Gaussian noise using deterministic DDIM sampling, and conclude that time series diffusion models exhibit reproducibility across all representation domains (see Appendix D).

## 5 CONCLUSION

We introduced WaveletDiff, a wavelet-space diffusion model that operates directly on wavelet coefficients with dedicated transformers for each decomposition level and cross-level attention mechanisms. Our approach captures multi-scale temporal patterns and preserves spectral characteristics through energy preservation training objectives. Extensive experiments across six diverse datasets demonstrate that WaveletDiff consistently outperforms state-of-the-art baselines across all evaluation metrics and sequence lengths, achieving discriminative scores and Context-FID scores that are $3\times$ smaller on average than the second-best baseline.

## THE USE OF LARGE LANGUAGE MODELS (LLMS)

LLMs assisted with drafting portions of the text, correcting spelling and grammatical errors, and improving clarity and style. In addition, LLMs were used in a limited capacity to assist with debugging during code development. All technical contributions, experimental design, mathematical formulations, and core insights are the original work of the authors. The authors take full responsibility for the final content, including any LLM-assisted text or code that was subsequently reviewed and validated.

## ETHICS STATEMENT

To the best of our knowledge, our work does not raise any ethical concerns. WaveletDiff is a time series generation method that operates on synthetic data generation for applications such as data augmentation, privacy preservation, and forecasting research. The datasets used in our experiments (ETTh1, ETTh2, Stocks, Exchange Rate, fMRI, and EEG) are publicly available research datasets that do not contain sensitive personal information. Our method does not involve human subjects, and the synthetic time series generation capability could potentially benefit privacy preservation by enabling the creation of synthetic datasets that maintain statistical properties while protecting individual privacy.

## REPRODUCIBILITY STATEMENT

We provide comprehensive implementation details to ensure reproducibility of our results. Model architecture specifications, hyperparameters, and training configurations are detailed in Section 3.2 and Appendix B.4. Dataset descriptions and preprocessing steps are provided in Section 4.1 and Appendix B.1. Evaluation metrics and experimental protocols are specified in Section 4.1 and Appendix B.2. Wavelet selection criteria and mother wavelet analysis are documented in Appendix B.3. Computational requirements and training procedures are outlined in Appendix B.5. All experimental settings, including cross-level attention mechanisms, energy preservation constraints, and ablation study configurations, are thoroughly described in the main paper and corresponding appendix sections to enable complete reproduction of our findings. Our code is available at https://anonymous.4open.science/r/WaveletDiff-27E9/.

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

# A  Mother Wavelet Families

Different wavelet families provide distinct characteristics of multi-scale decompositions through their specific filter coefficients $\{h_k\}$ and $\{g_k\}$ in the two-scale relations (Equation 1). In our experiments, we used five representative wavelet families, each satisfying different relationships between their order $p$ and filter length $F$. The vanishing moment properties and definitions presented here follow the PyWavelets framework implementation Lee et al. (2019b).

## A.1  Daubechies Wavelets (db)

Daubechies wavelets of order $p$ (e.g., db-$p$) provide orthogonality, compact support, and exactly $p$ vanishing moments for the wavelet function $\psi(t)$, with filter length $F = 2p$. The scaling function $\phi(t)$ has zero vanishing moments for orthogonal wavelets. The filter coefficients $\{h_k\}_{k=0}^{F-1}$ are derived from polynomial factorization to maximize regularity an maintain compact support.

For db2 (Daubechies-2 with $p = 2$, $F = 4$), the low-pass filter coefficients equal:

$$h_0 = \frac{1+\sqrt{3}}{4\sqrt{2}}, \quad h_1 = \frac{3+\sqrt{3}}{4\sqrt{2}}, \tag{15}$$

$$h_2 = \frac{3-\sqrt{3}}{4\sqrt{2}}, \quad h_3 = \frac{1-\sqrt{3}}{4\sqrt{2}}. \tag{16}$$

The high-pass coefficients satisfy $g_k = (-1)^k h_{F-1-k}$. The orthogonality and vanishing moment conditions ensure that

$$\sum_{k=0}^{F-1} h_k = \sqrt{2}, \tag{17}$$

$$\sum_{k=0}^{F-1} h_k h_{k+2m} = \delta_{m,0}, \tag{18}$$

$$\sum_{k=0}^{F-1} k^j h_k = 0 \quad \text{for } j = 1, \dots, p-1, \tag{19}$$

where $\delta_{m,0}$ is the Kronecker delta function, defined as:

$$\delta_{m,0} = \begin{cases} 1 & \text{if } m = 0 \\ 0 & \text{if } m \neq 0. \end{cases} \tag{20}$$

## A.2  Symlets (sym)

Symlets are modified Daubechies wavelets designed to improve symmetry while maintaining orthogonality and compact support. With a filter length $F = 2p$, Symlets have the same vanishing moment properties as Daubechies wavelets, i.e., $p$ vanishing moments for the wavelet function $\psi(t)$ and zero vanishing moments for the scaling function $\phi(t)$. They minimize an asymmetry measure $A$ that quantifies the deviation from perfect symmetry, namely

$$A = \sum_{k=0}^{F-1} k \cdot |h_k|^2 - \frac{F-1}{2} \sum_{k=0}^{F-1} |h_k|^2, \tag{21}$$

where the first term represents the weighted center of mass of the filter coefficients, while the second term represents the theoretical center for a perfectly symmetric filter. A lower value of $A$ indicates better symmetry.

For sym2 ($p = 2$, $F = 4$), the coefficients are optimized versions of db2 coefficients, satisfying the same orthogonality conditions but with improved phase linearity and near-symmetric properties for better temporal localization.

## A.3 COIFLETS (COIF)

Coiflets of order $p$ are designed with balanced vanishing moments for both the scaling $\phi(t)$ and wavelet function $\psi(t)$, with filter length $F = 6p$. The wavelet function $\psi(t)$ has $2p$ vanishing moments while the scaling function $\phi(t)$ has $2p-1$ vanishing moments, providing more balance for the moments when compared to Daubechies wavelets (in which case the scaling function has zero vanishing moments). The filter coefficients satisfy extended moment conditions of the form

$$\sum_{k=0}^{F-1} h_k = \sqrt{2}, \tag{22}$$

$$\sum_{k=0}^{F-1} k^j h_k = 0 \quad \text{for } j = 1, \ldots, 2p-1, \tag{23}$$

$$\sum_{k=0}^{F-1} g_k = 0, \tag{24}$$

$$\sum_{k=0}^{F-1} k^j g_k = 0 \quad \text{for } j = 1, \ldots, 2p. \tag{25}$$

For coif1 ($p = 1$, $F = 6$), the wavelet function has two vanishing moments and the scaling function has one vanishing moment. The six filter coefficients provide enhanced moment balancing between analysis and synthesis operations, with the scaling function having non-zero vanishing moments unlike orthogonal Daubechies wavelets.

## A.4 BIORTHOGONAL WAVELETS (BIOR)

Biorthogonal wavelets use different filters for decomposition and reconstruction, denoted as bior$p_r.p_d$ where $p_r$ and $p_d$ are the orders that determine the vanishing moment properties. For a general bior$p_r.p_d$ wavelet, we have the following:

- *Wavelet function $\psi(t)$*: $p_r$ vanishing moments;
- *Scaling function $\phi(t)$*: $p_d$ vanishing moments;
- *Filter lengths*: These depend on the specific bior$p_r.p_d$ configuration and are not given by simple formulas like those of other wavelet families.

For decomposition one uses a low-pass filter $\{h_k\}$ and a high-pass filter $\{g_k\}$, while for reconstruction one uses a low-pass filter $\{\tilde{h}_k\}$ (denoted with tilde) and a high-pass filter $\{\tilde{g}_k\}$. The tilde notation $\tilde{\ }$ indicates the dual (reconstruction) filters that are different from the primal (decomposition) filters.

For bior2.2 ($p_r = p_d = 2$), both decomposition and reconstruction wavelet functions have two vanishing moments, and both scaling functions have two vanishing moments, which result in perfect symmetry. The filters also satisfy the perfect reconstruction condition,

$$\sum_k h_k \tilde{h}_{k+2m} + g_k \tilde{g}_{k+2m} = \delta_{m,0} \tag{26}$$

In the z-domain, where $H(z)$, $G(z)$, $\tilde{H}(z)$, and $\tilde{G}(z)$ are the z-transforms of the respective filter sequences, the perfect reconstruction condition is succinctly summarized as

$$H(z)\tilde{H}(z^{-1}) + H(-z)\tilde{H}(-z^{-1}) = 2. \tag{27}$$

## A.5 REVERSE BIORTHOGONAL WAVELETS (RBIO)

Reverse biorthogonal wavelets (rbio$p_r.p_d$) interchange the decomposition and reconstruction filter roles compared to standard biorthogonal wavelets, according to:

$$h_k^{\text{rbio}} = \tilde{h}_k^{\text{bior}} \tag{28}$$

$$g_k^{\text{rbio}} = \tilde{g}_k^{\text{bior}} \tag{29}$$

For reverse biorthogonal wavelets, the vanishing moment assignment follows the same pattern as biorthogonal wavelets:

- *Wavelet function* $\psi(t)$ has $p_r$ vanishing moments.
- *Scaling function* $\phi(t)$ has $p_d$ vanishing moments.

For rbio2.2 ($p_r = p_d = 2$), both the wavelet function $\psi(t)$ and scaling function $\phi(t)$ have two vanishing moments each, maintaining symmetric properties.

The choice of wavelet family affects the sparsity and localization properties of the decomposition, with symmetric wavelets (bior, rbio) providing better phase preservation, while orthogonal wavelets (db, sym) ensure energy conservation through orthogonality.

## B  EXPERIMENTAL DETAILS

### B.1  DATASETS

Table 3 lists detailed properties of the datasets used in our experiments, and their repository links.

| Dataset | # of Samples | Dim | Source |
|---|---|---|---|
| ETTh1 | 17420 | 7 | https://github.com/zhouhaoyi/ETDataset |
| ETTh2 | 17420 | 7 | https://github.com/zhouhaoyi/ETDataset |
| Stocks | 3685 | 6 | https://finance.yahoo.com/quote/GOOG |
| Exchange Rate | 7588 | 8 | https://github.com/laiguokun/multivariate-time-series-data |
| fMRI | 10000 | 50 | https://www.fmrib.ox.ac.uk/datasets/netsim |
| EEG | 14980 | 14 | https://archive.ics.uci.edu/dataset/264/eeg+eye+state |

Table 3: Summary of the dataset types and statistics.

### B.2  METRICS

**Discriminative Score.** The discriminative score captures how difficult it is for a classifier to distinguish between real and generated samples. The score is measured by $|acc - 0.5|$, where acc is the classification accuracy. A score close to $0$ indicates that real and generated samples are indistinguishable to the classifier, while a score close to $0.5$ indicates they are very different. We follow the setup of TimeGAN Yoon et al. (2019) using a 2-layer GRU-based neural network as the classifier, trained with binary cross-entropy loss to distinguish between real (label=1) and synthetic (label=0) sequences.

**Predictive Score.** The predictive score captures how useful generated samples are for the forecasting task on real data. The score is measured by the mean absolute error (MAE) between predicted values and ground-truth values on test data. We follow TimeGAN Yoon et al. (2019) using a 2-layer GRU-based sequence predictor trained on synthetic data to predict the next time step features, evaluated on real sequences. Lower MAE values indicate better predictive utility of the generated samples.

**Context-FID.** Paul et al. (2022) The Fréchet Inception Distance (FID) measures the distance between two multivariate Gaussian distributions, i.e.,

$$\text{FID}(X, Y) = ||\mu_X - \mu_Y||^2 + \text{Tr}(\Sigma_X + \Sigma_Y - 2(\Sigma_X \Sigma_Y)^{1/2}), \tag{30}$$

where $\mu_X, \mu_Y$ are the means and $\Sigma_X, \Sigma_Y$ are the covariance matrices of the two distributions. Context-FID adapts this to time series by replacing the Inception-v3 features with time series features. We follow Diffusion-TS Yuan & Qiao (2024) using TS2Vec Yue et al. (2022) representations as the features. We extract embeddings from both real and generated sequences using a trained TS2Vec encoder, then compute FID in the embedding space. Lower Context-FID values indicate better distributional similarity.

**Correlational Score.** This metric assesses temporal dependencies by comparing cross-correlation matrices between real and generated data. For sequences with $D$ features, we compute the sample covariance matrix for each dataset, convert them to correlation matrices, and then measure the

average absolute difference across all feature pairs according to

$$\text{Correlational Score} = \frac{1}{10} \sum_{i=1}^{D} \sum_{j=1}^{D} |\rho_{i,j}^{real} - \rho_{i,j}^{generated}|, \tag{31}$$

where $\rho_{i,j}^{real}$ and $\rho_{i,j}^{generated}$ are the correlation coefficients between features $i$ and $j$ for real and generated data, respectively. Note that we follow the Diffusion-TS Yuan & Qiao (2024) setup using the factor $\frac{1}{10}$, although $\frac{1}{D^2}$ could provide better normalization across different feature dimensions. The former choice of normalization ensures direct comparability with prior work.

**DTW-JS Distance.** We propose a Dynamic Time Warping Jensen-Shannon Distance (DTW-JS distance) metric, which combines DTW's temporal alignment capabilities with Jensen-Shannon divergence for distributional comparison. DTW computes the optimal alignment distance between two time series sequences $x$ and $y$ by minimizing

$$\text{DTW}(x, y) = \min_{\pi} \sum_{(i,j) \in \pi} |x_i - y_j| \tag{32}$$

where $x$ and $y$ are two time series sequences, $\pi$ represents a warping path consisting of index pairs $(i, j)$ that map elements from sequence $x$ to sequence $y$, and the path must satisfy DTW constraints: monotonicity (indices only increase), continuity (no skipping), and boundary conditions (path starts at $(1, 1)$ and ends at $(|x|, |y|)$). The warping allows sequences to be stretched or compressed along the time axis to find the best alignment, enabling DTW to handle sequences of different lengths and account for temporal shifts or speed variations between similar patterns.

For our metric, we create a reference set $\mathcal{M}$ by randomly sampling from both real samples $\mathcal{R}$ and generated samples $\mathcal{G}$ (i.e., by taking the union of samples of these two sets, and ensuring that both sets have the same number of elements). For each sample $s$ in the real set $\mathcal{R}$ and generated set $\mathcal{G}$, we compute its mean DTW distance to all samples in the reference set:

$$d(s) = \frac{1}{|\mathcal{M}|} \sum_{r \in \mathcal{M}} \text{DTW}(s, r) \tag{33}$$

This creates two collections of mean DTW distances, which we histogram across different samples $s$ to form distance distributions $D_{\mathcal{R}}$ and $D_{\mathcal{G}}$ for real and generated samples, respectively. We then apply Jensen-Shannon divergence to compute the distance between the two distance distributions, i.e.,

$$\text{DTW-JS}(D_{\mathcal{R}}, D_{\mathcal{G}}) = \frac{1}{2} [\text{KL}(D_{\mathcal{R}} || D_M) + \text{KL}(D_{\mathcal{G}} || D_M)] \tag{34}$$

where $D_M = \frac{1}{2}(D_{\mathcal{R}} + D_{\mathcal{G}})$ is the mixture of the two distance distributions. This approach measures distributional similarity between real and generated samples while accounting for temporal alignment flexibility, providing a robust evaluation metric that captures both temporal structure and statistical properties.

### B.3 Wavelet Basis Function Analysis

The choice of mother wavelet significantly influences the multi-scale decomposition characteristics and subsequent series generation quality. Different wavelet families exhibit distinct properties in terms of orthogonality, compact support, and smoothness, which directly affect the sparsity and localization of coefficients in the wavelet domain. This choice becomes especially critical when dealing with diverse dataset characteristics, as different signal types require wavelets that can optimally capture their specific temporal-spectral patterns. As shown in Table 4, we systematically evaluated five representative wavelet families: Daubechies (db), Symlets (sym), Coiflets (coif), Biorthogonal (bior), and reverse Biorthogonal wavelets (rbio).

The results reveal dataset-specific preferred wavelets: Symlets works best for the Stocks dataset, likely due to their enhanced symmetry properties that better capture the near-symmetric fluctuations characteristic of financial time series. Coiflets demonstrate the best performance on the Exchange Rate dataset, benefiting from balanced vanishing moments for both scaling and wavelet functions, which effectively capture the smooth yet complex dynamics of currency fluctuations. For the remaining datasets (ETTh1, ETTh2, fMRI, EEG), Daubechies wavelets consistently provide the best

overall performance. This ability to adapt the wavelet basis to match dataset characteristics represents an important advantage of WaveletDiff over frequency-domain approaches, which are constrained to use fixed Fourier basis functions regardless of the underlying signal properties, limiting their capacity to optimally represent diverse temporal patterns across different domains.

| Metrics | Wavelet | ETTh1 | ETTh2 | Stocks | Exchange Rate | fMRI | EEG |
|---|---|---|---|---|---|---|---|
| Discriminative Score (Lower the Better) | db | **0.005±.005** | **0.008±.007** | 0.013±.007 | **0.004±.001** | 0.175±.071 | **0.006±.008** |
| | sym | 0.023±.005 | 0.023±.005 | **0.005±.004** | 0.011±.009 | 0.196±.066 | 0.007±.003 |
| | coif | 0.025±.010 | 0.031±.004 | 0.017±.009 | 0.073±.010 | **0.087±.077** | 0.014±.014 |
| | bior | 0.022±.008 | 0.033±.002 | 0.012±.004 | 0.051±.012 | 0.273±.007 | 0.008±.004 |
| | rbio | 0.057±.009 | 0.074±.008 | 0.010±.008 | 0.094±.010 | 0.129±.127 | 0.017±.009 |
| Predictive Score (Lower the Better) | db | 0.119±.002 | 0.106±.004 | **0.037±.000** | 0.037±.002 | **0.100±.000** | **0.000±.000** |
| | sym | 0.117±.004 | 0.107±.003 | **0.037±.000** | **0.035±.003** | **0.100±.000** | **0.000±.000** |
| | coif | **0.115±.005** | 0.106±.004 | **0.037±.000** | 0.036±.003 | **0.100±.000** | **0.000±.000** |
| | bior | 0.122±.002 | 0.109±.004 | **0.037±.000** | 0.037±.001 | **0.100±.000** | **0.000±.000** |
| | rbio | 0.121±.004 | **0.104±.001** | **0.037±.000** | 0.036±.001 | **0.100±.000** | **0.000±.000** |
| Context-FID Score (Lower the Better) | db | **0.020±.001** | **0.023±.002** | 0.024±.004 | **0.006±.000** | 0.104±.003 | **0.006±.000** |
| | sym | 0.052±.004 | 0.051±.006 | 0.018±.002 | 0.009±.001 | 0.122±.007 | 0.011±.001 |
| | coif | 0.079±.008 | 0.069±.009 | 0.018±.003 | 0.108±.008 | 0.104±.006 | **0.006±.000** |
| | bior | 0.049±.003 | 0.156±.016 | **0.016±.001** | 0.088±.013 | 0.119±.004 | **0.006±.001** |
| | rbio | 0.161±.005 | 0.225±.041 | **0.016±.002** | 0.175±.027 | 0.176±.008 | 0.007±.001 |
| Correlational Score (Lower the Better) | db | 0.043±.008 | 0.083±.016 | 0.006±.003 | **0.060±.020** | 1.073±.005 | 1.811±.963 |
| | sym | 0.055±.008 | 0.073±.025 | 0.005±.003 | 0.066±.012 | 1.172±.048 | 2.164±.533 |
| | coif | **0.036±.006** | **0.064±.011** | 0.007±.004 | 0.167±.032 | 1.177±.031 | 1.971±.969 |
| | bior | 0.048±.011 | 0.094±.009 | 0.005±.003 | 0.137±.020 | 1.147±.033 | 3.034±.759 |
| | rbio | 0.051±.008 | 0.099±.026 | **0.003±.004** | 0.161±.018 | 1.402±.034 | 1.959±.707 |
| DTW-JS Distance (Lower the Better) | db | **0.101±.016** | **0.064±.014** | 0.121±.013 | **0.121±.029** | 0.199±.043 | 0.055±.011 |
| | sym | 0.123±.009 | 0.067±.023 | **0.106±.027** | 0.132±.017 | 0.283±.035 | **0.049±.015** |
| | coif | 0.104±.015 | 0.086±.015 | 0.115±.016 | 0.157±.022 | **0.191±.011** | 0.062±.017 |
| | bior | 0.117±.013 | 0.095±.019 | 0.109±.009 | 0.129±.033 | 0.280±.011 | 0.050±.009 |
| | rbio | 0.110±.021 | 0.086±.033 | 0.119±.024 | 0.144±.037 | 0.464±.021 | 0.068±.022 |

Table 4: Mother Wavelet Selection

## B.4 TRAINING CONFIGURATION DETAILS

We provide training configuration information needed for reproducibility. The WaveletDiff model uses an embedding dimension of 256 with 8 attention heads across 8 transformer layers, a time embedding dimension of 128, and dropout rate of 0.1. The approximation level transformer uses twice as many embedding dimensions (512) and twice the number of layers (16) to capture the critically important low-frequency information.

The diffusion process employs 1000 timesteps with an exponential noise schedule. The exponential noise schedule is of the form

$$\beta_t = \beta_{start} + (\beta_{end} - \beta_{start}) \cdot \left(1 - e^{-\gamma \cdot t}\right) \tag{35}$$

where $\beta_{start} = 0.0001$, $\beta_{end} = 0.02$, $\gamma = 2.0$ is the exponential decay rate, $t \in [0, 1]$ is the normalized timestep, and $T = 1000$ is the total number of timesteps. The coefficient-weighted loss strategy assigns an approximation coefficient weight of 2.0 to emphasize low-frequency components. We train for 5000 epochs with batch sizes 512.

For optimization, we use the AdamW optimizer with initial learning rate $2 \times 10^{-4}$ and weight decay $1 \times 10^{-5}$. We employ a one-cycle learning rate schedule Smith & Topin (2018) with cosine annealing strategy. The learning rate follows a two-phase schedule: Linear warm-up for the first 30% of training, then cosine annealing for the remaining 70%:

$$lr(e) = \begin{cases} lr_{base} + (lr_{max} - lr_{base}) \times \frac{e}{p \times E}, & \text{if } e \le p \times E; \\ lr_{final} + (lr_{max} - lr_{final}) \times \left(\frac{1 + \cos\left(\pi \times \frac{e - p \times E}{(1-p) \times E}\right)}{2}\right), & \text{if } e > p \times E. \end{cases} \tag{36}$$

where $e$ is the current epoch, $lr_{base} = 4 \times 10^{-5}$, $lr_{max} = 1 \times 10^{-3}$, $lr_{final} = 4 \times 10^{-9}$, $p = 0.3$, and $E = 5000$ total epochs.

## B.5 Computational Setup and Training Time Analysis

WaveletDiff experiments were conducted on a single NVIDIA H100 GPU with 80GB memory using PyTorch 2.7.1 with CUDA 11.8, with training performed for 5000 epochs. Our WaveletDiff model contains approximately 63M trainable parameters. FourierDiffusion baseline results were obtained on a Tesla T4 GPU due to hardware constraints, following their original configuration and training settings. We note that the difference in GPU hardware was unavoidable due to High-Performance Computing (HPC) resource availability and the PyTorch Lightning deployment used for WaveletD-iff, and is not selected to give any unfair advantage to our model. Tables 5 and 6 present the training times for WaveletDiff and FourierDiffusion across different datasets and sequence lengths, respectively. Despite the large model and long training, WaveletDiff completes training in a relatively short time using only a single GPU, indicating that compute time is clearly not a bottleneck.

Table 5: Training times (hours:minutes:seconds) for WaveletDiff across different datasets and sequence lengths on single NVIDIA H100 GPU.

| sequence length | ETTh1 | ETTh2 | Stocks | Exchange Rate | fMRI | EEG |
|---|---|---|---|---|---|---|
| 24 | 3:45:54 | 3:36:34 | 1:07:05 | 1:36:59 | 2:40:13 | 3:22:06 |
| 32 | 3:45:33 | 3:38:23 | 1:02:49 | 1:46:32 | 2:42:41 | 3:24:07 |
| 64 | 3:24:46 | 4:19:04 | 1:12:34 | 2:03:44 | 3:15:34 | 4:13:58 |
| 128 | 5:06:03 | 6:27:03 | 1:33:53 | 2:59:33 | 4:32:52 | 6:01:14 |

Table 6: Training times (hours:minutes:seconds) for FourierDiffusion across different datasets and sequence lengths on single NVIDIA Tesla T4 GPU.

| sequence length | ETTh1 | ETTh2 | Stocks | Exchange Rate | fMRI | EEG |
|---|---|---|---|---|---|---|
| 24 | 1:09:51 | 1:19:01 | 1:57:23 | 2:21:39 | 2:21:38 | 2:39:03 |
| 32 | 1:38:56 | 1:39:13 | 2:10:41 | 2:30:54 | 2:27:28 | 3:47:09 |
| 64 | 4:49:37 | 4:49:39 | 3:14:13 | 2:31:45 | 3:58:26 | 5:21:55 |
| 128 | 10:14:05 | 10:14:04 | 6:35:18 | 3:46:43 | 6:29:26 | 7:16:49 |

| Metrics | Methods | ETTh1 | ETTh2 | Stocks | Exchange Rate | fMRI | EEG |
|---|---|---|---|---|---|---|---|
| Discriminative Score (Lower the Better) | WaveletDiff | **0.005±.005** | **0.008±.007** | **0.005±.004** | **0.004±.001** | **0.087±.077** | **0.006±.008** |
| | coefficient prediction | 0.017±.013 | 0.040±.030 | 0.027±.014 | 0.059±.010 | 0.277±.012 | 0.020±.013 |
| | w/o cross attention | 0.055±.054 | 0.028±.017 | 0.016±.012 | 0.052±.011 | 0.179±.016 | 0.006±.003 |
| | cosine noise scheduler | 0.024±.014 | 0.021±.019 | 0.094±.013 | 0.012±.005 | 0.112±.037 | 0.500±.000 |
| | DDIM sampling | 0.135±.094 | 0.017±.005 | 0.010±.003 | 0.006±.004 | 0.494±.006 | 0.024±.017 |
| Predictive Score (Lower the Better) | WaveletDiff | 0.119±.002 | **0.106±.004** | **0.037±.000** | 0.037±.002 | **0.100±.000** | **0.000±.000** |
| | coefficient prediction | 0.120±.003 | 0.111±.003 | **0.037±.000** | 0.038±.003 | **0.100±.000** | **0.000±.000** |
| | w/o cross attention | 0.119±.002 | **0.106±.002** | **0.037±.000** | 0.037±.001 | **0.100±.000** | **0.000±.000** |
| | cosine noise scheduler | 0.119±.003 | 0.107±.003 | **0.037±.000** | 0.037±.002 | 0.103±.000 | 0.172±.239 |
| | DDIM sampling | **0.118±.005** | 0.106±.004 | **0.037±.000** | 0.036±.002 | 0.101±.000 | **0.000±.000** |
| Context-FID Score (Lower the Better) | WaveletDiff | **0.020±.001** | **0.023±.002** | 0.018±.002 | **0.006±.000** | 0.104±.006 | **0.006±.000** |
| | coefficient prediction | 0.027±.003 | 0.059±.004 | 0.053±.007 | 0.085±.008 | 0.131±.010 | 0.009±.001 |
| | w/o cross attention | 0.118±.006 | 0.044±.004 | 0.039±.006 | 0.042±.004 | 0.170±.008 | 0.008±.001 |
| | cosine noise scheduler | 0.125±.005 | 0.050±.009 | 0.116±.021 | 0.011±.002 | 0.329±.022 | 44246±3637 |
| | DDIM sampling | 0.135±.006 | 0.049±.004 | **0.008±.002** | **0.006±.000** | 2.394±.043 | 0.009±.001 |
| Correlational Score (Lower the Better) | WaveletDiff | **0.043±.008** | 0.083±.016 | **0.005±.003** | **0.060±.020** | **1.177±.031** | 1.811±.963 |
| | coefficient prediction | 0.056±.010 | 0.097±.029 | 0.006±.003 | 0.182±.019 | 2.005±.051 | 2.650±.314 |
| | w/o cross attention | 0.046±.016 | **0.078±.016** | 0.006±.004 | 0.092±.035 | 1.755±.062 | 1.993±.738 |
| | cosine noise scheduler | 0.050±.014 | 0.105±.026 | 0.029±.010 | 0.070±.009 | 2.175±.071 | 1.487±.579 |
| | DDIM sampling | 0.057±.011 | 0.094±.018 | **0.005±.003** | 0.080±.031 | 2.113±.044 | **0.844±.150** |
| DTW-JS Distance (Lower the Better) | WaveletDiff | **0.101±.016** | **0.064±.014** | **0.106±.027** | **0.121±.029** | **0.191±.011** | 0.055±.011 |
| | coefficient prediction | 0.115±.017 | 0.103±.030 | 0.133±.021 | 0.140±.024 | 0.204±.029 | 0.071±.019 |
| | w/o cross attention | 0.116±.029 | 0.085±.013 | 0.127±.020 | 0.133±.025 | 0.348±.009 | 0.064±.007 |
| | cosine noise scheduler | 0.102±.018 | 0.075±.007 | 0.127±.021 | 0.134±.010 | 0.357±.033 | 0.693±.000 |
| | DDIM sampling | 0.112±.020 | 0.074±.018 | 0.113±.011 | 0.122±.022 | 0.693±.000 | **0.040±.014** |

Table 7: Ablation study results for key WaveletDiff architectural components.

## C    VISUALIZATION

### C.1    T-SNE AND DATA DISTRIBUTION ON SHORT SEQUENCE GENERATION

We present the t-SNE and data distribution visualization of short sequence generation on ETTh2, Exchange Rate, fMRI, and EEG dataset in Figure 5, 6, 7, and 8. The results demonstrate that WaveletDiff consistently outperforms baseline methods in capturing the underlying data distributions across all datasets.

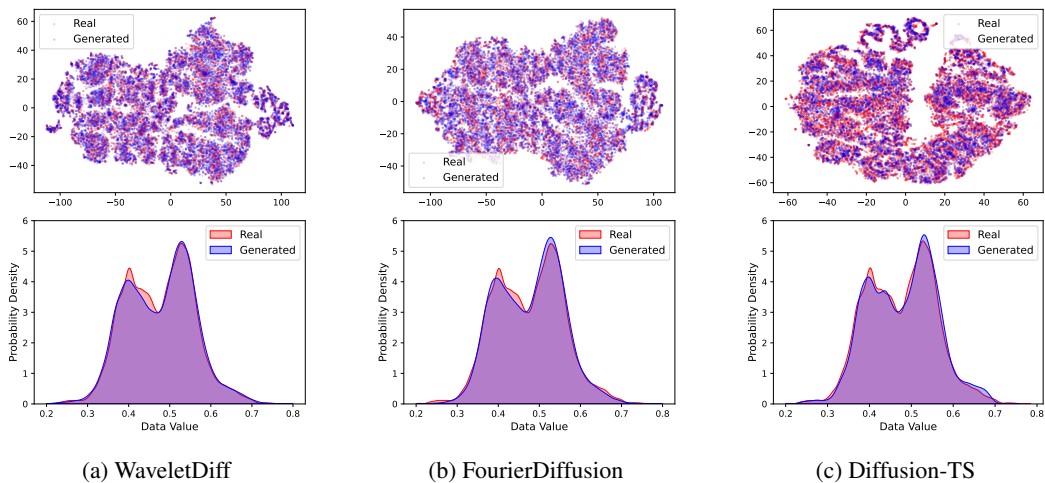

(a) WaveletDiff                (b) FourierDiffusion                (c) Diffusion-TS

Figure 5: t-SNE visualization and probability distribution of data values on ETTh2 dataset.

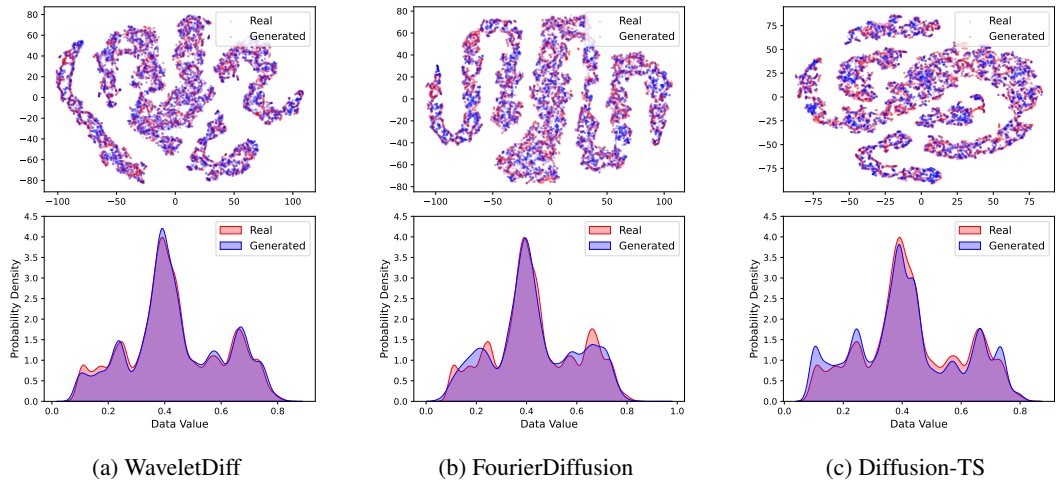

(a) WaveletDiff                (b) FourierDiffusion                (c) Diffusion-TS

Figure 6: t-SNE visualization and probability distribution of data values on Exchange Rate dataset.

## D    DIFFUSION MODEL REPRODUCIBILITY ANALYSIS

Inspired by recent work on reproducibility in diffusion models for images Zhang et al. (2024b); Li et al. (2024); Kadkhodaie et al. (2024), we examine whether this phenomenon extends to time series generation. Specifically, we train pairs of models with slightly different architectures or configurations, then generate samples from the same fixed Gaussian noise input using deterministic DDIM sampling. For each pair of generated sequences $(x_1, x_2)$, we compute their similarity using dynamic time warping (DTW) distance. To quantify reproducibility, we follow Zhang et al. (2024b)

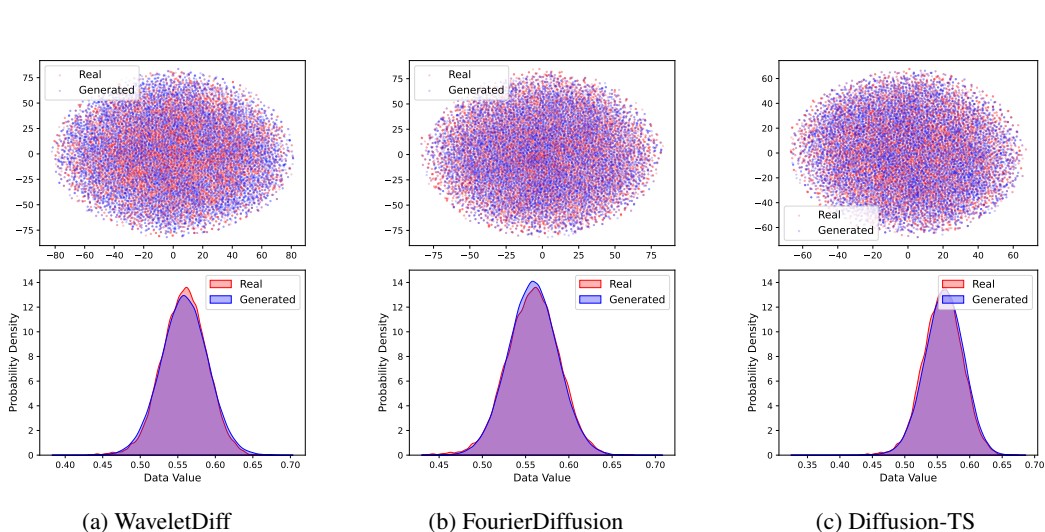

(a) WaveletDiff     (b) FourierDiffusion     (c) Diffusion-TS

Figure 7: t-SNE visualization and probability distribution of data values on fMRI dataset.

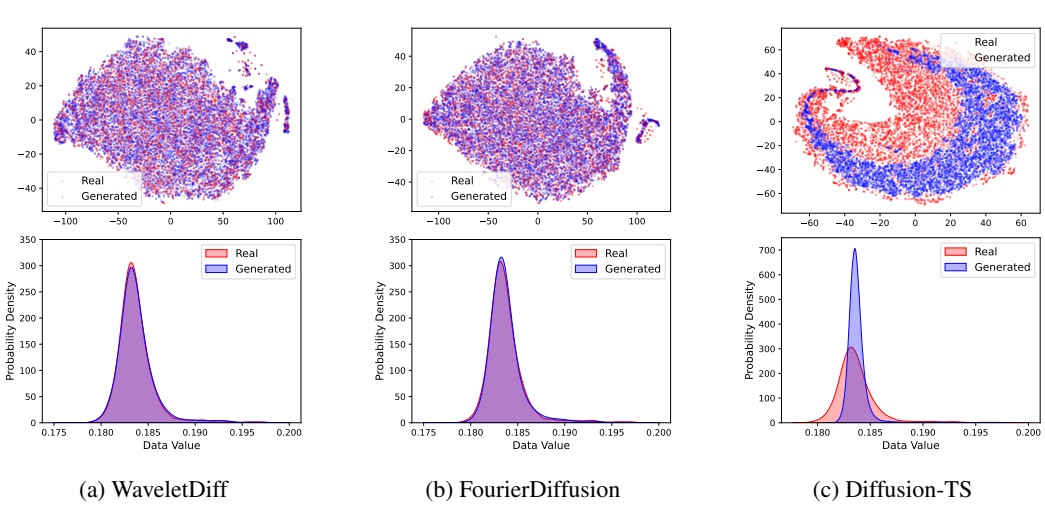

(a) WaveletDiff     (b) FourierDiffusion     (c) Diffusion-TS

Figure 8: t-SNE visualization and probability distribution of data values on EEG dataset.

and define the RP score as

$$\text{RP score} := \mathbb{P}\big(\text{DTW}(x_1, x_2) < \overline{\text{DTW}}_{\text{rand}}\big), \tag{37}$$

where $\overline{\text{DTW}}_{\text{rand}}$ denotes the average DTW distance between randomly chosen sequence pairs generated by the two models. Thus, the RP score measures the probability that two models produce more similar samples from the same noise than would be expected by chance. An RP score greater than 0.5 indicates reproducibility.

Unlike image generation, which commonly uses U-Net architectures for comparison, time series generation employs diverse architectures. We examine the RP score across different model variations for both WaveletDiff and FourierDiffusion. Table 8 demonstrates that reproducibility exists for time series data regardless of the representation domain (wavelet, time, or Fourier). To the best of our knowledge, we are the first to examine the reproducibility phenomenon specifically for time series generation.

| Datasets | Model 1 | Model 2 | RP score |
|---|---|---|---|
| Stocks | WaveletDiff | WaveletDiff w/o cross-attention | 1.0 |
| | | WaveletDiff + cosine noise scheduler | 0.66 |
| | FourierDiffusion | FourierDiffusion on time domain | 0.665 |
| | | FourierDiffusion using LSTM score model | 0.805 |
| Exchange Rate | WaveletDiff | WaveletDiff w/o cross-attention | 0.999 |
| | | WaveletDiff + cosine noise scheduler | 0.619 |
| | FourierDiffusion | FourierDiffusion on time domain | 0.610 |
| | | FourierDiffusion using LSTM score model | 0.945 |

Table 8: Reproducibility scores for different model variations demonstrate that time series diffusion models exhibit reproducibility across architectural changes and representation domains.

### D.1 IMPACT OF ARCHITECTURAL VARIATIONS

We evaluate how architectural modifications affect reproducibility by removing the cross-attention module from WaveletDiff. Figures 9 and 10 show that the model maintains strong reproducibility despite this significant architectural change, with generated sequences from identical noise exhibiting nearly identical patterns.

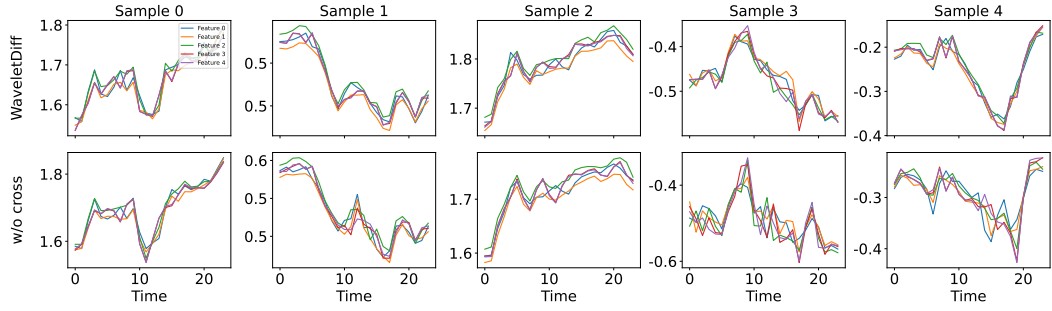

Figure 9: Reproducibility comparison on Stocks dataset using identical initial noise (volume feature excluded for clarity).

### D.2 IMPACT OF MOTHER WAVELET SELECTION

We further investigate how different mother wavelet choices affect reproducibility. As shown in Figures 11 and 12, Daubechies and Symlets wavelets demonstrate high reproducibility, while Coiflets, Biorthogonal, and Reverse Biorthogonal wavelets also exhibit good consistency. This indicates that wavelet choice significantly affects the learned distribution, with similar wavelet families (e.g., orthogonal wavelets like Daubechies and Symlets) producing more comparable results than dissimilar families.

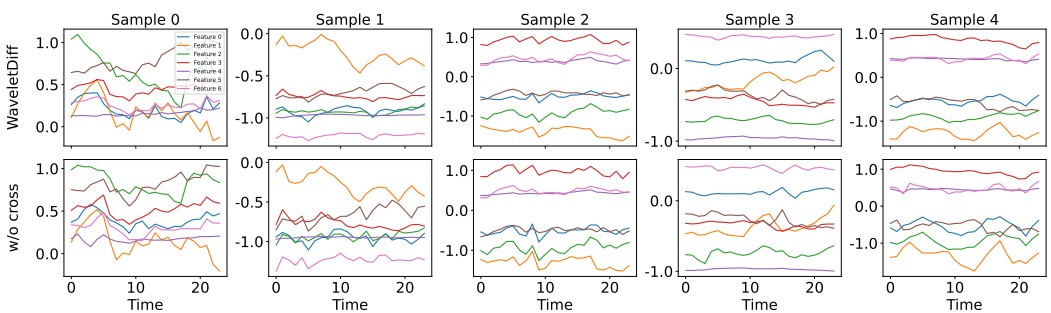

Figure 10: Reproducibility comparison on Exchange Rate dataset using identical initial noise.

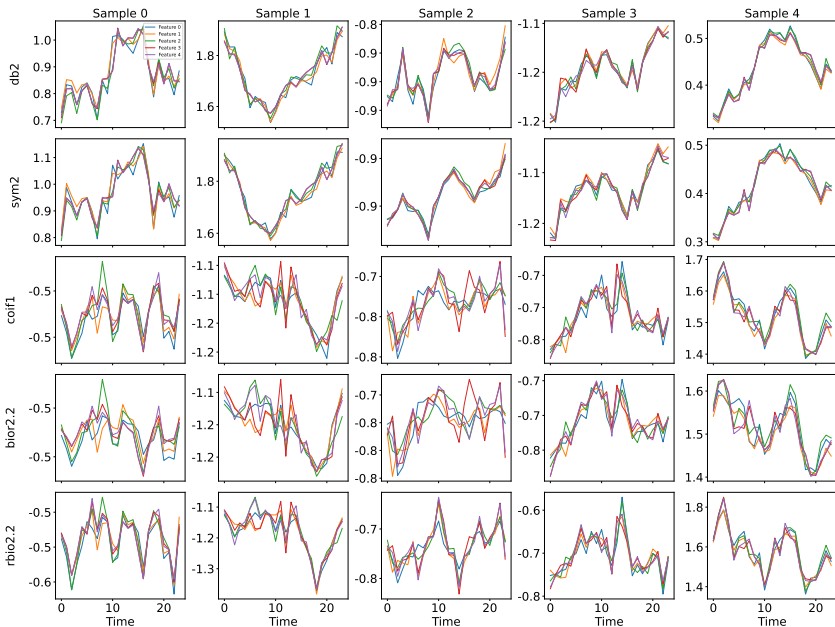

Figure 11: Wavelet family comparison on Stocks dataset demonstrating varying reproducibility across different mother wavelets.

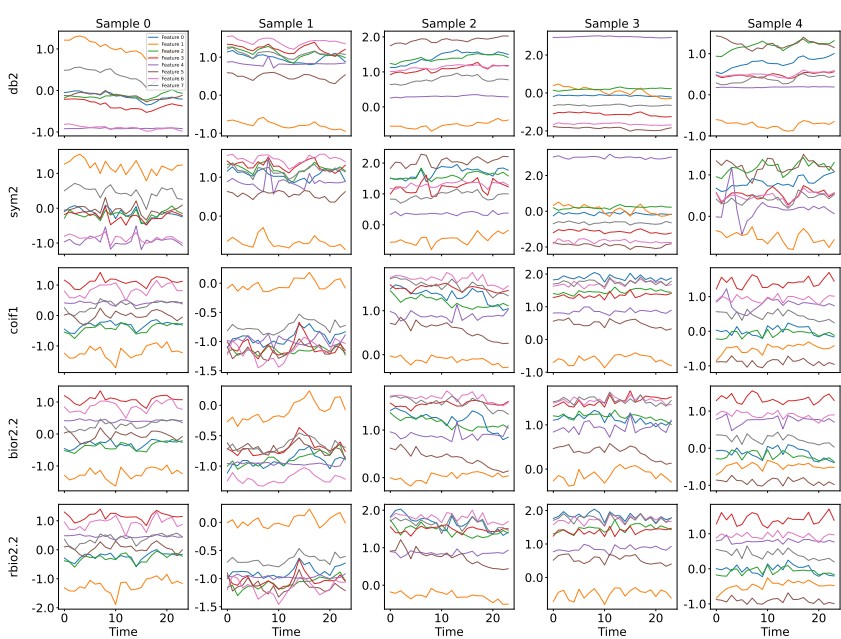

Figure 12: Wavelet family comparison on Exchange Rate dataset demonstrating varying reproducibility across mother wavelets.

