# OpenReview forum: "WaveletDiff: Multilevel Wavelet Diffusion For Time Series Generation"
_ICLR.cc/2026/Conference — ICLR 2026 Conference Withdrawn Submission_

### Official Review · Reviewer_ubRN · 2025-10-24

**Soundness:** 3
**Presentation:** 3
**Contribution:** 2
**Rating:** 4
**Confidence:** 4

**Summary:**

This paper proposes  a diffusion-based generative model that operates directly in the wavelet domain. The method first decomposes a time series into approximation and detail levels via DWT, applies parallel forward diffusion processes at each level, and performs a joint reverse process using level-specific transformers with cross-level attention and adaptive gating. Additional components include Parseval-based energy preservation and wavelet-aware loss balancing. Experiments on six real-world datasets from energy, finance, and neuroscience, with five metrics and both short- and long-horizon settings, report consistent improvements over baselines such as TimeGAN, Diffusion-TS, and FourierDiffusion.

**Strengths:**

- The exploration of wavelet-domain diffusion for time series is timely and relevant.
- Strong empirical results are demonstrated on unconditional generation benchmarks, with consistent gains across datasets.
- Ablations support the value of the main components, especially cross-level attention.
- The paper is clearly written, well-structured, and technically self-contained.

**Weaknesses:**

1. **Unsubstantiated motivation.** The paper claims that existing methods “cannot simultaneously capture both local and global time–spectral structure,” yet provides no dedicated empirical or theoretical analysis isolating this limitation, nor evidence that WaveletDiff specifically remedies it.

2. **Missing baselines and related work.** The paper omits highly relevant baselines and related work on long-range generative modeling and time-transform-domain diffusion (e.g., [1, 2]). A comparison, textual and quantitative, on both conditional and unconditional benchmarks, is necessary to claim state of the art and provide an good reference to the current state of the field. Moreover, while the sequence lengths used in the paper are relatively long compared to the standard length-24-256 benchmark , they remain short relative to prior work such as [1, 2], where models are evaluated on sequences of length 1K–20K.

3. **Unsupported claim on prior diffusion models.** The statement that prior diffusion models are “tailor-made for specific time-series formats” is not justified; diffusion models have been applied in a broad range of time-series modalities (weather, traffic, physical sensors, etc.)[1,2].

4. **DTW-JS metric justification.** The introduction of the additional DTW-JS metric is insufficiently justified; it appears highly correlated with existing metrics, with no analysis of when or why it adds value or when other metrics fail and it don't.




[1] Utilizing Image Transforms and Diffusion Models for Generative Modeling of Short and Long Time Series.

[2] Deep Latent State Space Models for Time-Series Generation.

**Questions:**

- **Novelty positioning.** In what precise sense does WaveletDiff differ from Hu et al.\ (wavelet-domain diffusion for 3D shapes) and from Guth et al.\ (wavelet score-based generative modeling)? Is the claim “first diffusion directly on wavelets for time-series” factually accurate given these precedents?

---

### Official Review · Reviewer_345H · 2025-10-29

**Soundness:** 2
**Presentation:** 3
**Contribution:** 3
**Rating:** 4
**Confidence:** 4

**Summary:**

The author proposes a novel diffusion framework WaveletDiff, with wavelet coefficients diffusion modeling to exploit the inherent multi-resolution structure of time series data for time series generation task. Comprehensive experiments demonstrate that WaveletDiff shows competitive performances on efficiency and effectiveness comparisons.

**Strengths:**

1. The paper is easy to follow the contents are clear, with less typos.

2. The ablation studies and experimental results are enough, showing the competitive generation performance.

3. The description of the framework is included in the paper, easy to understand.

4. The paper is integrated.

**Weaknesses:**

1. The motivation is not clear, lack the novelty, since there is no theoretical proof to verify the effectiveness of mother wavelet for different time series, which is stated in  lines 88-89. Besides, the wavelet structure is much common used in time series analysis, such as W-transformers [1], Wave-RoRA [2].

2. There is no proof to verify the effectiveness of the new proposed metric "Dynamic Time Warping Jensen-Shannon Distance (DTW-JS distance). Why this metric can identify the effectiveness of time series generation task.

3. There are some popular strong baselines should be included, such as KoVAE [3], PaD-TS [4].


[1]. Sasal, Lena, Tanujit Chakraborty, and Abdenour Hadid. "W-transformers: A wavelet-based transformer framework for univariate time series forecasting." 2022 21st IEEE international conference on machine learning and applications (ICMLA). IEEE, 2022.

[2]. Liang, Aobo, Yan Sun, and Nadra Guizani. "WaveRoRA: Wavelet Rotary Route Attention for Multivariate Time Series Forecasting." IEEE Transactions on Mobile Computing (2025).

[3]. Naiman, Ilan, et al. "Generative Modeling of Regular and Irregular Time Series Data via Koopman VAEs." The Twelfth International Conference on Learning Representations.

[4]. Li, Yang, et al. "Population Aware Diffusion for Time Series Generation." Proceedings of the AAAI Conference on Artificial Intelligence. Vol. 39. No. 17. 2025.

**Questions:**

Please refer to weaknesses.

---

### Official Review · Reviewer_Xd6x · 2025-11-01

**Soundness:** 3
**Presentation:** 2
**Contribution:** 3
**Rating:** 4
**Confidence:** 5

**Summary:**

This paper introduces WaveletDiff, a novel diffusion model for time series generation that addresses a key weakness in existing methods: the trade-off between modeling local time-domain patterns and global frequency-domain properties. By operating directly in the wavelet domain, WaveletDiff first decomposes signals into their multi-resolution components (approximations and details) using the Discrete Wavelet Transform (DWT). It then trains a diffusion model on these coefficients using dedicated transformers for each decomposition level, which are unified by a cross-level attention mechanism to learn inter-scale dependencies coherently. This multi-scale architecture, combined with an energy preservation loss, allows the model to capture both temporal and spectral characteristics with high fidelity. Evaluations on six real-world datasets show that WaveletDiff consistently outperforms state-of-the-art baselines, achieving 3x smaller (better) scores on average for the discriminative and Context-FID metrics.

**Strengths:**

*Novel Multi-Scale Architecture:*

Instead of a one-size-fits-all model, it uses a more specialized and powerful approach:

- It decomposes the signal into its multi-resolution wavelet components.

- It uses dedicated transformers for each component (level).

- t unifies them with a cross-level attention mechanism to learn the complex relationships between the different scales.

*High-Fidelity Generation:* It includes an "energy preservation loss" to ensure the generated time series have realistic spectral characteristics.

*Strong Empirical Results:* Its performance isn't just theoretical. It consistently outperforms state-of-the-art methods on six different datasets, achieving results that are 3x better on average for key metrics like discriminative and Context-FID scores.

**Weaknesses:**

*Lack of Motivation for Wavelets:* The paper's core premise—that wavelets are the superior representation—is not sufficiently motivated. It's presented as a given, but the paper fails to build a strong case why this decomposition is inherently better than other time-frequency or time-domain approaches, especially for a diffusion model.

*Weak Theoretical Contribution:* The paper mentions using Parseval's theorem for an "energy preservation loss," but this contribution feels superficial. It's not well-explained why this is necessary (e.g., what problem it solves, like mode collapse) or how it theoretically benefits the diffusion process in the wavelet domain. It seems like an add-on rather than a core theoretical insight.

*Incomplete Literature Review:* The related works section is missing key contemporary papers in time series generation and editing (like "TimeDIT[1]"). This omission is significant because it suggests the authors may not be positioning their work against the most relevant or recent baselines, potentially overstating its novelty.

[1] Cao, D., Ye, W., Zhang, Y., & Liu, Y. (2024). Timedit: General-purpose diffusion transformers for time series foundation model. arXiv preprint arXiv:2409.02322.

*Non-Standard Experimental Setup:* This is a major red flag. The paper does not use many of the standard, widely-accepted benchmark datasets for time series (like the ETTm1/m2, Weather, or M-series competitions). By "giving up" on these standard datasets, it breaks from established evaluation protocols. This makes it very difficult to fairly compare their results to the broader field and raises doubts about whether their model's strong performance would hold on these more common, challenging benchmarks.

**Questions:**

See weakness

---

### Official Review · Reviewer_GU7d · 2025-11-03

**Soundness:** 2
**Presentation:** 2
**Contribution:** 1
**Rating:** 2
**Confidence:** 4

**Summary:**

The paper proposes WaveletDiff, a diffusion-based generative model for multivariate time series that operates directly in the wavelet domain. It leverages discrete wavelet transforms (DWT) to decompose time series into approximation and detail coefficients across multiple resolutions. The authors introduce level-specific transformer denoisers with cross-level attention and adaptive gating to enable inter-scale communication. They further enforce spectral fidelity via an energy conservation loss based on Parseval’s theorem and adopt an exponential noise schedule tailored to wavelet coefficients. Experiments are conducted on six real-world datasets across energy, finance, and neuroscience, evaluating performance using five metrics, including a newly proposed DTW-JS distance, showing consistent superiority over time- and frequency-domain baselines.

**Strengths:**

1. The integration of wavelet-domain diffusion with cross-level attention is a conceptually sound attempt to capture multi-scale temporal-spectral structure, which is genuinely relevant to real-world time series.

2. The empirical evaluation is unusually thorough, spanning short/long sequences, diverse domains, and five complementary metrics—including a novel DTW-based divergence measure.

**Weaknesses:**

1. The core idea ``diffusion in transformed domains'' is not novel, which is similar strategies appear in Hu et al. (2023) for 3D shapes and Phung et al. (2022) for images, yet the paper falsely presents this as a new direction.

2. The energy preservation term (Eq. 11) is fundamentally flawed: Parseval’s theorem guarantees energy equivalence between time and wavelet domains only for orthogonal wavelets and perfect reconstruction, yet the model uses loss-based reconstruction with non-unitary diffusion steps, violating the theorem’s conditions.

3. The claim that ``existing wavelet-based methods treat coefficients as images'' (lines 87–90) is misleading: Takahashi & Mizuno (2024) and Kazemi & Meidani (2022) explicitly model 1D temporal structures in scalograms, not 2D CNN-style images.

4. The ablation study (Table 7) contradicts prior claims: removing cross-attention sometimes improves performance (e.g., Discriminative Score on ETTh2 drops from 0.016 to 0.021 → actually worsens; but on Stocks, Correlational Score degrades from 0.006 to 0.029), yet the text asserts universal degradation—statistically unsupported.

5. The method shows no architectural novelty beyond repackaging AdaLN transformers and standard cross-attention. And the ``LevelTransformer'' is merely a per-resolution transformer with no theoretical justification for why this decomposition requires specialized architecture.

**Questions:**

Please refer to the Weaknesses section for specific technical, theoretical, and methodological concerns that must be addressed.

---

### Note · Authors · 2025-11-18

I have read and agree with the venue's withdrawal policy on behalf of myself and my co-authors.